# GENERATIVE MODEL-ENHANCED HUMAN MOTION PREDICTION

## ABSTRACT

The task of predicting human motion is complicated by the natural heterogeneity and compositionality of actions, necessitating robustness to distributional shifts as far as out-of-distribution (OoD). Here we formulate a new OoD benchmark based on the Human3.6M and CMU motion capture datasets, and introduce a hybrid framework for hardening discriminative architectures to OoD failure by augmenting them with a generative model. When applied to current state-of-the-art discriminative models, we show that the proposed approach improves OoD robustness without sacrificing in-distribution performance, and can theoretically facilitate model interpretability. We suggest human motion predictors ought to be constructed with OoD challenges in mind, and provide an extensible general framework for hardening diverse discriminative architectures to extreme distributional shift.

## 1 INTRODUCTION

Human motion is naturally intelligible as a time-varying graph of connected joints constrained by locomotor anatomy and physiology. Its prediction allows the anticipation of actions with applications across healthcare (Geertsema et al., 2018; Kakar et al., 2005), physical rehabilitation and training (Chang et al., 2012; Webster & Celik, 2014), robotics (Koppula & Saxena, 2013b;a; Gui et al., 2018b), navigation (Paden et al., 2016; Alahi et al., 2016; Bhattacharyya et al., 2018; Wang et al., 2019), manufacture (Švec et al., 2014), entertainment (Shirai et al., 2007; Rofougaran et al., 2018; Lau & Chan, 2008), and security (Kim & Paik, 2010; Ma et al., 2018).

The favoured approach to predicting movements over time has been purely inductive, relying on the history of a specific class of movement to predict its future. For example, state space models (Koller & Friedman, 2009) enjoyed early success for simple, common or cyclic motions (Taylor et al., 2007; Sutskever et al., 2009; Lehrmann et al., 2014). The range, diversity and complexity of human motion has encouraged a shift to more expressive, deep neural network architectures (Fragkiadaki et al., 2015; Butepage et al., 2017; Martinez et al., 2017; Li et al., 2018; Aksan et al., 2019; Mao et al., 2019; Li et al., 2020b; Cai et al., 2020), but still within a simple inductive framework.

This approach would be adequate were actions both sharply distinct and highly stereotyped. But their complex, compositional nature means that within one category of action the kinematics may vary substantially, while between two categories they may barely differ. Moreover, few *real-world* tasks restrict the plausible repertoire to a small number of classes–distinct or otherwise–that could be explicitly learnt. Rather, any action may be drawn from a great diversity of possibilities–both kinematic and teleological–that shape the characteristics of the underlying movements. This has two crucial implications. First, any modelling approach that lacks awareness of the full space of motion possibilities will be vulnerable to poor generalisation and brittle performance in the face of kinematic anomalies. Second, the very notion of *In-Distribution* (ID) testing becomes moot, for the relations between different actions and their kinematic signatures are plausibly determinable only across the entire domain of action. A test here arguably needs to be *Out-of-Distribution* (OoD) if it is to be considered a robust test at all.

These considerations are amplified by the nature of real-world applications of kinematic modelling, such as anticipating arbitrary *deviations* from expected motor behaviour early enough for an automatic intervention to mitigate them. Most urgent in the domain of autonomous driving (Bhattacharyya et al., 2018; Wang et al., 2019), such *safety* concerns are of the highest importance, and

are best addressed within the fundamental modelling framework. Indeed, Amodei et al. (2016) cites the ability to recognize our own ignorance as a safety mechanism that must be a core component in safe AI. Nonetheless, to our knowledge, current predictive models of human kinematics neither quantify OoD performance nor are designed with it in mind. There is therefore a need for two *frameworks*, applicable across the domain of action modelling: one for *hardening* a predictive model to anomalous cases, and another for *quantifying* OoD performance with established benchmark datasets. General frameworks are here desirable in preference to new models, for the field is evolving so rapidly greater impact can be achieved by introducing mechanisms that can be applied to a breadth of candidate architectures, even if they are demonstrated in only a subset. Our approach here is founded on combining a latent variable generative model with a standard predictive model, illustrated with the current state-of-the-art discriminative architecture (Mao et al., 2019; Wei et al., 2020), a strategy that has produced state-of-the-art in the medical imaging domain Myronenko (2018). Our aim is to achieve robust performance within a realistic, low-volume, high-heterogeneity data regime by providing a general mechanism for enhancing a discriminative architecture with a generative model.

In short, our contributions to the problem of achieving robustness to distributional shift in human motion prediction are as follows:

1. We provide a framework to benchmark OoD performance on the most widely used open-source motion capture datasets: Human3.6M (Ionescu et al., 2013), and CMU-Mocap[1], and evaluate state-of-the-art models on it.

2. We present a framework for hardening deep feed-forward models to OoD samples. We show that the hardened models are fast to train, and exhibit substantially improved OoD performance with minimal impact on ID performance.

We begin section 2 with a brief review of human motion prediction with deep neural networks, and of OoD generalisation using generative models. In section 3, we define a framework for benchmarking OoD performance using open-source multi-action datasets. We introduce in section 4 the discriminative models that we harden using a generative branch to achieve a state-of-the-art (SOTA) OoD benchmark. We then turn in section 5 to the architecture of the generative model and the overall objective function. Section 6 presents our experiments and results. We conclude in section 7 with a summary of our results, current limitations, and caveats, and future directions for developing robust and reliable OoD performance and a quantifiable awareness of unfamiliar behaviour.

## 2 RELATED WORK

**Deep-network based human motion prediction.** Historically, sequence-to-sequence prediction using Recurrent Neural Networks (RNNs) have been the de facto standard for human motion prediction (Fragkiadaki et al., 2015; Jain et al., 2016; Martinez et al., 2017; Pavllo et al., 2018; Gui et al., 2018a; Guo & Choi, 2019; Gopalakrishnan et al., 2019; Li et al., 2020b). Currently, the SOTA is dominated by feed forward models (Butepage et al., 2017; Li et al., 2018; Mao et al., 2019; Wei et al., 2020). These are inherently faster and easier to train than RNNs. The jury is still out, however, on the optimal way to handle temporality for human motion prediction. Meanwhile, recent trends have overwhelmingly shown that graph-based approaches are an effective means to encode the spatial dependencies between joints (Mao et al., 2019; Wei et al., 2020), or sets of joints (Li et al., 2020b). In this study, we consider the SOTA models that have graph-based approaches with a feed forward mechanism as presented by (Mao et al., 2019), and the subsequent extension which leverages motion attention, Wei et al. (2020). We show that these may be augmented to improve robustness to OoD samples.

**Generative models for Out-of-Distribution prediction and detection.** Despite the power of deep neural networks for prediction in complex domains (LeCun et al., 2015), they face several challenges that limits their suitability for safety-critical applications. Amodei et al. (2016) list *robustness to distributional shift* as one of the five major challenges to AI safety. Deep generative models, have been used extensively for detection of OoD inputs and have been shown to generalise

---

[1]t http://mocap.cs.cmu.edu/

well in such scenarios (Hendrycks & Gimpel, 2016; Liang et al., 2017; Hendrycks et al., 2018). While recent work has showed some failures in simple OoD detection using density estimates from deep generative models (Nalisnick et al., 2018; Daxberger & Hernández-Lobato, 2019), they remain a prime candidate for anomaly detection (Kendall & Gal, 2017; Grathwohl et al., 2019; Daxberger & Hernández-Lobato, 2019).

Myronenko (2018) use a Variational Autoencoder (VAE) (Kingma & Welling, 2013) to regularise an encoder-decoder architecture with the specific aim of better generalisation. By simultaneously using the encoder as the recognition model of the VAE, the model is encouraged to base its segmentations on a complete picture of the data, rather than on a reductive representation that is more likely to be fitted to the training data. Furthermore, the original loss and the VAE's loss are combined as a weighted sum such that the discriminator's objective still dominates. Further work may also reveal useful interpretability of behaviour (via visualisation of the latent space as in Bourached & Nachev (2019)), generation of novel motion (Motegi et al., 2018), or reconstruction of missing joints as in Chen et al. (2015).

## 3 QUANTIFYING OUT-OF-DISTRIBUTION PERFORMANCE OF HUMAN MOTION PREDICTORS

Even a very compact representation of the human body such as OpenPose's 17 joint parameterisation Cao et al. (2018) explodes to unmanageable complexity when a temporal dimension is introduced of the scale and granularity necessary to distinguish between different kinds of action: typically many seconds, sampled at hundredths of a second. Moreover, though there are anatomical and physiological constraints on the space of licit joint configurations, and their trajectories, the repertoire of possibility remains vast, and the kinematic demarcations of teleologically different actions remain indistinct. Thus, no practically obtainable dataset may realistically represent the possible distance between instances. To simulate OoD data we first need ID data that can be varied in its quantity and heterogeneity, closely replicating cases where a particular kinematic morphology may be rare, and therefore undersampled, and cases where kinematic morphologies are both highly variable within a defined class and similar across classes. Such replication needs to accentuate the challenging aspects of each scenario.

We therefore propose to evaluate OoD performance where only a single action, drawn from a single action distribution, is available for training and hyperparameter search, and testing is carried out on the remaining classes. In appendix A, to show that the action categories we have chosen can be distinguished at the time scales on which our trajectories are encoded, we train a simple classifier and show it separates the selected ID action from the others with high accuracy (100% precision and recall for the CMU dataset). Performance over the remaining set of actions may thus be considered OoD.

## 4 BACKGROUND

Here we describe the current SOTA model proposed by Mao et al. (2019) (GCN). We then describe the extension by Wei et al. (2020) (attention-GCN) which antecedes the GCN prediction model with motion attention.

### 4.1 PROBLEM FORMULATION

We are given a motion sequence $\mathbf{X}_{1:N} = (\mathbf{x}_1, \mathbf{x}_2, \mathbf{x}_3, \cdots, \mathbf{x}_N)$ consisting of $N$ consecutive human poses, where $\mathbf{x}_i \in \mathbb{R}^K$, with $K$ the number of parameters describing each pose. The goal is to predict the poses $\mathbf{X}_{N+1:N+T}$ for the subsequent $T$ time steps.

### 4.2 DCT-BASED TEMPORAL ENCODING

The input is transformed using Discrete Cosine Transformations (DCT). In this way each resulting coefficient encodes information of the entire sequence at a particular temporal frequency. Furthermore, the option to remove high or low frequencies is provided. Given a joint, $k$, the position of $k$ over $N$ time steps is given by the trajectory vector: $\mathbf{x}_k = [x_{k,1}, \ldots, x_{k,N}]$ where we convert to a

DCT vector of the form: $\mathbf{C}_k = [C_{k,1}, \ldots, C_{k,N}]$ where $C_{k,l}$ represents the lth DCT coefficient. For $\delta_{l1} \in \mathbb{R}^N = [1, 0, \cdots, 0]$, these coefficients may be computed as

$$C_{k,l} = \sqrt{\frac{2}{N}} \sum_{n=1}^{N} x_{k,n} \frac{1}{\sqrt{1 + \delta_{l1}}} \cos\left(\frac{\pi}{2N}(2n-1)(l-1)\right). \tag{1}$$

If no frequencies are cropped, the DCT is invertible via the Inverse Discrete Cosine Transform (IDCT):

$$x_{k,l} = \sqrt{\frac{2}{N}} \sum_{l=1}^{N} C_{k,l} \frac{1}{\sqrt{1 + \delta_{l1}}} \cos\left(\frac{\pi}{2N}(2n-1)(l-1)\right). \tag{2}$$

Mao et al. use the DCT transform with a graph convolutional network architecture to predict the output sequence. This is achieved by having an equal length input-output sequence, where the input is the DCT transformation of $\mathbf{x_k} = [x_{k,1}, \ldots, x_{k,N}, x_{k,N+1}, \ldots, x_{k,N+T}]$, here $[x_{k,1}, \ldots, x_{k,N}]$ is the observed sequence and $[x_{k,N+1}, \ldots, x_{k,N+T}]$ are replicas of $x_{k,N}$ (ie $x_{k,n} = x_{k,N}$ for $n \geq N$). The target is now simply the ground truth $\mathbf{x_k}$.

### 4.3 Graph Convolutional Network

Suppose $\mathbf{C} \in \mathbb{R}^{K \times (N+T)}$ is defined on a graph with $k$ nodes and $N + T$ dimensions, then we define a graph convolutional network to respect this structure. First we define a Graph Convolutional Layer (GCL) that, as input, takes the activation of the previous layer ($\mathbf{A}^{[l-1]}$), where $l$ is the current layer.

$$GCL(\mathbf{A}^{[l-1]}) = \mathbf{S}\mathbf{A}^{[l-1]}\mathbf{W} + \mathbf{b} \tag{3}$$

where $\mathbf{A}^{[0]} = \mathbf{C} \in \mathbb{R}^{K \times (N+T)}$, and $\mathbf{S} \in \mathbb{R}^{K \times K}$ is a layer-specific learnable normalised graph laplacian that represents connections between joints, $\mathbf{W} \in \mathbb{R}^{n^{[l-1]} \times n^{[l]}}$ are the learnable inter-layer weightings and $\mathbf{b} \in \mathbb{R}^{n^{[l]}}$ are the learnable biases where $n^{[l]}$ are the number of hidden units in layer $l$.

### 4.4 Network Structure and Loss

The network consists of 12 Graph Convolutional Blocks (GCBs), each containing 2 GCLs with skip (or residual) connections, see figure 7. Additionally, there is one GCL at the beginning of the network, and one at the end. $n^{[l]} = 256$, for each layer, $l$. There is one final skip connection from the DCT inputs to the DCT outputs, which greatly reduces train time. The model has around 2.6M parameters. Hyperbolic tangent functions are used as the activation function. Batch normalisation is applied before each activation.

The outputs are converted back to their original coordinate system using the IDCT (equation 2) to be compared to the ground truth. The loss used for joint angles is the average $l_1$ distance between the ground-truth joint angles, and the predicted ones. Thus, the joint angle loss is:

$$\ell_a = \frac{1}{K(N+T)} \sum_{n=1}^{N+T} \sum_{k=1}^{K} |\hat{x}_{k,n} - x_{k,n}| \tag{4}$$

where $\hat{x}_{k,n}$ is the predicted $k^{th}$ joint at timestep $n$ and $x_{k,n}$ is the corresponding ground truth.

This is separately trained on 3D joint coordinate prediction making use of the Mean Per Joint Position Error (MPJPE), as proposed in Ionescu et al. (2013) and used in Mao et al. (2019); Wei et al. (2020). This is defined, for each training example, as

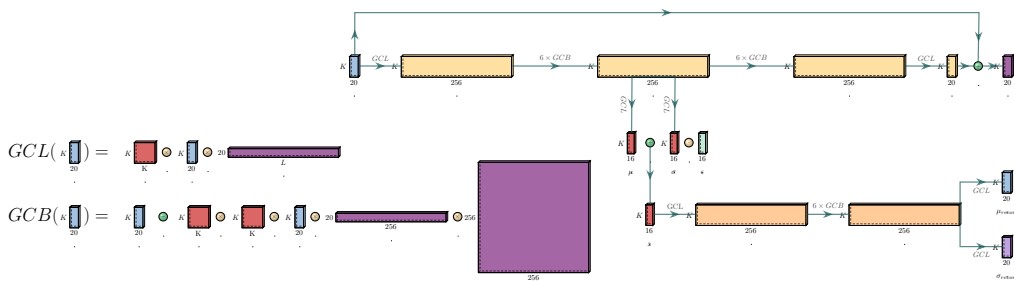

Figure 1: GCN network architecture with VAE branch. Here $n_z = 16$ is the number of latent variables per joint.

$$\ell_m = \frac{1}{J(N+T)} \sum_{n=1}^{N+T} \sum_{j=1}^{J} \|\hat{\mathbf{p}}_{j,n} - \mathbf{p}_{j,n}\|^2 \tag{5}$$

where $\hat{\mathbf{p}}_{j,n} \in \mathbb{R}^3$ denotes the predicted jth joint position in frame $n$. And $\mathbf{p}_{j,n}$ is the corresponding ground truth, while J is the number of joints in the skeleton.

### 4.5 MOTION ATTENTION EXTENSION

Wei et al. (2020) extend this model by summing multiple DCT transformations from different sections of the motion history with weightings learned via an attention mechanism. For this extension, the above model (the GCN) along with the anteceding motion attention is trained end-to-end. We refer to this as the attention-GCN.

## 5 OUR APPROACH

Myronenko (2018) augment an encoder-decoder discriminative model by using the encoder as a recognition model for a Variational Autoencoder (VAE), (Kingma & Welling, 2013; Rezende et al., 2014). Myronenko (2018) show this to be a very effective regulariser. Here, we also use a VAE, but for conjugacy with the discriminator, we use graph convolutional layers in the decoder. This can be compared to the Variational Graph Autoencoder (VGAE), proposed by Kipf & Welling (2016). However, Kipf & Welling's application is a link prediction task in citation networks and thus it is desired to model only connectivity in the latent space. Here we model connectivity, position, and temporal frequency. To reflect this distinction, the layers immediately before, and after, the latent space are fully connected creating a homogenous latent space.

The generative model sets a precedence for information that can be modelled causally, while leaving elements of the discriminative machinery, such as skip connections, to capture correlations that remain useful for prediction but are not necessarily persuant to the objective of the generative model. In addition to performing the role of regularisation in general, we show that we gain robustness to distributional shift across similar, but different, actions that are likely to share generative properties. The architecture may be considered with the visual aid in figure 1.

### 5.1 VARIATIONAL AUTOENCODER (VAE) BRANCH AND LOSS

Here we define the first 6 GCB blocks as our VAE recognition model, with a latent variable $\mathbf{z} \in \mathbb{R}^{K \times n_z} = N(\mu_{\mathbf{z}}, \sigma_{\mathbf{z}})$, where $\mu_{\mathbf{z}} \in \mathbb{R}^{K \times n_z}, \sigma_{\mathbf{z}} \in \mathbb{R}^{K \times n_z}$. $n_z = 8$, or 32 depending on training stability.

The KL divergence between the latent space distribution and a spherical Gaussian $N(0, \mathbf{I})$ is given by:

$$\ell_l = KL(q(\mathbf{Z}|\mathbf{C})||q(\mathbf{Z})) = \frac{1}{2} \sum_1^{n_z} \left( \mu_{\mathbf{z}}^2 + \sigma_{\mathbf{z}}^2 - 1 - log((\sigma_{\mathbf{z}})^2) \right). \tag{6}$$

The decoder part of the VAE has the same structure as the discriminative branch; 6 GCBs. We parametrise the output neurons as $\mu \in \mathbb{R}^{K \times (N+T)}$, and $log(\sigma^2) \in \mathbb{R}^{K \times (N+T)}$. We can now model the reconstruction of inputs as samples of a maximum likelihood of a Gaussian distribution which constitutes the second term of the negative Variational Lower Bound (VLB) of the VAE:

$$\ell_G = log(p(\mathbf{C}|\mathbf{Z})) = -\frac{1}{2} \sum_{n=1}^{N+T} \sum_{l=1}^{K} \left( log(\sigma_{k,l}^2) + log(2\pi) + \frac{|C_{k,l} - \mu_{k,l}|^2}{e^{log(\sigma_{k,l}^2)}} \right), \tag{7}$$

where $C_{k,l}$ are the DCT coefficients of the ground truth.

## 5.2 TRAINING

We train the entire network together with the additional of the negative VLB:

$$\ell = \underbrace{\frac{1}{(N+T)K} \sum_{n=1}^{N+T} \sum_{k=1}^{K} |\hat{x}_{k,n} - x_{k,n}|}_{\text{Discriminitive loss}} - \lambda \underbrace{(\ell_G - \ell_l)}_{\text{VLB}}. \tag{8}$$

Here $\lambda$ is a hyperparameter of the model. The overall network is $\approx 3.4M$ parameters. The number of parameters varies slightly as per the number of joints, K, since this is reflected in the size of the graph in each layer ($k = 48$ for H3.6M, $K = 64$ for CMU joint angles, and $K = J = 75$ for CMU Cartesian coordinates). Furthermore, once trained, the generative model is not required for prediction and hence for this purpose is as compact as the original models.

## 6 EXPERIMENTS

### 6.1 DATASETS AND EXPERIMENTAL SETUP

**Human3.6M (H3.6M)** The H3.6M dataset (Ionescu et al., 2011; 2013), so called as it contains a selection of 3.6 million 3D human poses and corresponding images, consists of seven actors each performing 15 actions, such as walking, eating, discussion, sitting, and talking on the phone. Martinez et al. (2017); Mao et al. (2019); Li et al. (2020b) all follow the same training and evaluation procedure: training their motion prediction model on 6 (5 for train and 1 for cross-validation) of the actors, for each action, and evaluate metrics on the final actor, subject 5. For easy comparison to these ID baselines, we maintain the same train; cross-validation; and test splits. However, we use the single, most well-defined action (see appendix A), *walking*, for train and cross-validation, and we report test error on all the remaining actions from subject 5. In this way we conduct all parameter selection based on ID performance.

**CMU motion capture** (CMU-mocap) The CMU dataset consists of 5 general classes of actions. Similarly to (Li et al., 2018; 2020a; Mao et al., 2019) we use 8 detailed actions from these classes: 'basketball', 'basketball signal', 'directing traffic' 'jumping, 'running', 'soccer', 'walking', and 'window washing'. We use two representations, a 64-dimensional vector that gives an exponential map representation (Grassia, 1998) of the joint angle, and a 75-dimensional vector that gives the 3D Cartesian coordinates of 25 joints. We do not tune any hyperparameters on this dataset and use only a train and test set with the same split as is common in the literature (Martinez et al., 2017; Mao et al., 2019).

**Model configuration**  We implemented the model in PyTorch (Paszke et al., 2017) using the ADAM optimiser (Kingma & Ba, 2014). The learning rate was set to 0.0005 for all experiments where, unlike Mao et al. (2019); Wei et al. (2020), we did not decay the learning rate as it was hypothesised that the dynamic relationship between the discriminative and generative loss would make this redundant. The batch size was 16. For numerical stability, gradients were clipped to a maximum $\ell2$-norm of 1 and $log(\hat{\sigma}^2)$ and values were clamped between -20 and 3. Code for all experiments is available at the following anonymized link: https://anonymous.4open.science/r/11a7a2b5-da13-43f8-80de-51e526913dd2/

| milliseconds | Walking (ID) | | | | Eating (OoD) | | | | Smoking (OoD) | | | | Average (of 14 for OoD) | | | |
|---|---|---|---|---|---|---|---|---|---|---|---|---|---|---|---|---|
| | 80 | 160 | 320 | 400 | 80 | 160 | 320 | 400 | 80 | 160 | 320 | 400 | 80 | 160 | 320 | 400 |
| GCN (OoD) | **0.22** | **0.37** | 0.60 | 0.65 | 0.22 | 0.38 | 0.65 | 0.79 | **0.28** | 0.55 | 1.08 | 1.10 | **0.38** | 0.69 | 1.09 | 1.27 |
| Std Dev | 0.001 | 0.008 | . 0.008 | 0.01 | 0.003 | 0.01 | 0.03 | 0.04 | 0.01 | 0.01 | 0.02 | 0.02 | 0.007 | 0.02 | 0.04 | 0.04 |
| ours (OoD) | 0.23 | **0.37** | **0.59** | **0.64** | **0.21** | **0.37** | **0.59** | **0.72** | **0.28** | **0.54** | **1.01** | **0.99** | **0.38** | **0.68** | **1.07** | **1.21** |
| Std Dev | 0.003 | 0.004 | 0.03 | 0.03 | 0.008 | 0.01 | 0.03 | 0.04 | 0.005 | 0.01 | 0.01 | 0.02 | **0.006** | **0.01** | **0.01** | **0.02** |

Table 1: Short-term prediction of Eucildean distance between predicted and ground truth joint angles on H3.6M. Each experiment conducted 3 times. We report the mean and standard deviation. Note that we have lower variance in results for ours. Full table in appendix, table 6.

| milliseconds | Walking | | Eating | | Smoking | | Discussion | | Average | |
|---|---|---|---|---|---|---|---|---|---|---|
| | 560 | 1000 | 560 | 1000 | 560 | 1000 | 560 | 1000 | 560 | 1000 |
| GCN (OoD) | 0.80 | 0.80 | **0.89** | 1.20 | 1.26 | 1.85 | 1.45 | **1.88** | 1.10 | 1.43 |
| ours (OoD) | **0.66** | **0.72** | 0.90 | **1.19** | **1.17** | **1.78** | **1.44** | 1.90 | **1.04** | **1.40** |

Table 2: Long-term prediction of Eucildean distance between predicted and ground truth joint angles on H3.6M.

**Baseline comparison**  Both Mao et al. (2019) (GCN), and Wei et al. (2020) (attention-GCN) use this same Graph Convolutional Network (GCN) architecture with DCT inputs. In particular, Wei et al. (2020) increase the amount of history accounted for by the GCN by adding a motion attention mechanism to weight the DCT coefficients from different sections of the history prior to being inputted to the GCN. We compare against both of these baselines on OoD actions. For attention-GCN we leave the attention mechanism preceding the GCN unchanged such that the generative branch of the model is reconstructing the weighted DCT inputs to the GCN, and the whole network is end-to-end differentiable.

**Hyperparameter search**  Since a new term has been introduced to the loss function, it was necessary to determine a sensible weighting between the discriminative and generative models. In Myronenko (2018), this weighting was arbitrarily set to 0.1. It is natural that the optimum value here will relate to the other regularisation parameters in the model. Thus, we conducted random hyperparameter search for $p_{drop}$ and $\lambda$ in the ranges $p_{drop} = [0, 0.5]$ on a linear scale, and $\lambda = [10, 0.00001]$ on a logarithmic scale. For fair comparison we also conducted hyperparameter search on GCN, for values of the dropout probability ($p_{drop}$) between 0.1 and 0.9. For each model, 25 experiments were run and the optimum values were selected on the lowest ID validation error. The hyperparameter search was conducted only for the GCN model on short-term predictions for the H3.6M dataset and used for all future experiments hence demonstrating generalisability of the architecture.

## 6.2 RESULTS

| milliseconds | Basketball (ID) | | | | | Basketball Signal (OoD) | | | | | Average (of 7 for OoD) | | | | |
|---|---|---|---|---|---|---|---|---|---|---|---|---|---|---|---|
| | 80 | 160 | 320 | 400 | 1000 | 80 | 160 | 320 | 400 | 1000 | 80 | 160 | 320 | 400 | 1000 |
| GCN | **0.40** | 0.67 | **1.11** | **1.25** | **1.63** | **0.27** | **0.55** | **1.14** | **1.42** | 2.18 | 0.36 | 0.65 | 1.41 | 1.49 | 2.17 |
| ours | **0.40** | **0.66** | 1.12 | 1.29 | 1.76 | 0.28 | 0.57 | 1.15 | 1.43 | **2.07** | **0.34** | **0.62** | **1.35** | **1.41** | **2.10** |

Table 3: Eucildean distance between predicted and ground truth joint angles on CMU. Full table in appendix, table 7.

Consistent with the literature we report short-term ($< 500ms$) and long-term ($> 500ms$) predictions. In comparison to GCN, we take short term history into account (10 frames, $400ms$) for both

|  | Basketball | | | | | Basketball Signal | | | | | Average (of 7 for OoD) | | | | |
|---|---|---|---|---|---|---|---|---|---|---|---|---|---|---|---|
| milliseconds | 80 | 160 | 320 | 400 | 1000 | 80 | 160 | 320 | 400 | 1000 | 80 | 160 | 320 | 400 | 1000 |
| GCN (OoD) | **15.7** | **28.9** | **54.1** | **65.4** | 108.4 | 14.4 | 30.4 | 63.5 | 78.7 | 114.8 | **20.0** | 43.8 | 86.3 | 105.8 | 169.2 |
| ours (OoD) | 16.0 | 30.0 | 54.5 | 65.5 | **98.1** | **12.8** | **26.0** | **53.7** | **67.6** | **103.2** | 21.6 | **42.3** | **84.2** | **103.8** | **164.3** |

Table 4: Mean Joint Per Position Error (MPJPE) between predicted and ground truth 3D Cartesian coordinates of joints on CMU. Full table in appendix, table 8.

datasets to predict both short- and long-term motion. In comparison to attention-GCN, we take long term history (50 frames, 2 seconds) to predict the next 10 frames, and predict futher into the future by recursively applying the predictions as input to the model as in Wei et al. (2020). In this way a single short term prediction model may produce long term predictions.

We use Euclidean distance between the predicted and ground-truth joint angles for the Euler angle representation. For 3D joint coordinate representation we use the MPJPE as used for training (equation 5). Table 1 reports the joint angle error for the short term predictions on the H3.6M dataset. Here we found the optimum hyperparameters to be $p_{drop} = 0.5$ for GCN, and $\lambda = 0.003$, with $p_{drop} = 0.3$ for our augmentation of GCN. The latter of which was used for all future experiments, where for our augmentation of attention-GCN we removed dropout altogether. On average, our model performs convincingly better both ID and OoD. Here the generative branch works well as both a regulariser for small datasets and by creating robustness to distributional shifts. We see similar and consistent results for long-term predictions in table 2.

From tables 3 and 4, we can see that the superior OoD performance generalises to the CMU dataset with the same hyperparameter settings with a similar trend of the difference being larger for longer predictions for both joint angles and 3D joint coordinates. For each of these experiments $n_z = 8$.

Table 5, shows that the effectiveness of the generative branch generalises to the very recent motion attention architecture. For attention-GCN we used $n_z = 32$. Here, interestingly short term predictions are poor but long term predictions are consistently better. This supports our assertion that information relevant to generative mechanisms are more intrinsic to the causal model and thus, here, when the predicted output is recursively used, more useful information is available for the future predictions.

|  | Walking (ID) | | | | Eating (OoD) | | | | Smoking (OoD) | | | | Average (of 14 for OoD) | | | |
|---|---|---|---|---|---|---|---|---|---|---|---|---|---|---|---|---|
| milliseconds | 560 | 720 | 880 | 1000 | 560 | 720 | 880 | 1000 | 560 | 720 | 880 | 1000 | 560 | 720 | 880 | 1000 |
| att-GCN (OoD) | **55.4** | **60.5** | **65.2** | **68.7** | 87.6 | 103.6 | 113.2 | 120.3 | 81.7 | 93.7 | 102.9 | 108.7 | **112.1** | 129.6 | 140.3 | 147.8 |
| ours (OoD) | 58.7 | 60.6 | 65.5 | 69.1 | **81.7** | **94.4** | **102.7** | **109.3** | **80.6** | **89.9** | **99.2** | **104.1** | 113.1 | **127.7** | **137.9** | **145.3** |

Table 5: Long-term prediction of 3D joint positions on H3.6M. Here, ours is also trained with the attention-GCN model. Full table in appendix, table 9.

# 7 CONCLUSION

We draw attention to the need for robustness to distributional shifts in predicting human motion, and propose a framework for its evaluation based on major open source datasets. We demonstrate that state-of-the-art discriminative architectures can be hardened to extreme distributional shifts by augmentation with a generative model, combining low in-distribution predictive error with maximal generalisability. The introduction of a surveyable latent space further provides a mechanism for model perspicuity and interpretability, and explicit estimates of uncertainty facilitate the detection of anomalies: both characteristics are of substantial value in emerging applications of motion prediction, such as autonomous driving, where safety is paramount. Our investigation argues for wider use of generative models in behavioural modelling, and shows it can be done with minimal or no performance penalty, within hybrid architectures of potentially diverse constitution.

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

## APPENDIX

The appendix consists of 4 parts. We provide a brief summary of each section below.

Appendix A: we provide results from our experimentation to determine the optimum way of defining separable distributions on the H3.6M, and the CMU datasets.

Appendix B: we provide the full results tables which are shown in part in the main text.

Appendix C: we inspect the generative model by examining its latent space and use it to consider the role that the gnerative model plays in learning as well as possible directions of future work.

Appendix D: we provide larger diagrams of the architecture of the augmented GCN.

## A    DEFINING *Out-of-Distribution* (OOD)

Here we describe in more detail the empirical motivation for our definition of *Out-of-Distribution* (OoD) on the H3.6M and CMU datasets.

Figure 2 shows the distribution of actions for the h3.6M and CMU datasets. We want our ID data to be small in quantity, and narrow in domain. Since this dataset is labelled by action we are provided with a natural choice of distribution being one of these actions. Moreover, it is desirable that the action be quantifiably distinct from the other actions.

To determine which action supports these properties we train a simple classifier to determine which action is most easily distinguished from the others based on the DCT inputs: $DCT(\vec{x}_k) = DCT([x_{k,1}, \ldots, x_{k,N}, x_{k,N+1}, \ldots, x_{k,N+T}])$ where $x_{k,n} = x_{k,N}$ for $n \geq N$. We make no assumption on the architecture that would be optimum to determine the separation, and so use a simple fully connected model with 4 layers. Layer 1: *input dimensions* $\times$ 1024, layer 2: $1024 \times 512$, layer 3: $512 \times 128$, layer 4: $128 \times 15$ (or $128 \times 8$ for CMU). Where the final layer uses a softmax to predict the class label. Cross entropy is used as a loss function on these logits during training. We used ReLU activations with a dropout probability of 0.5.

We trained this model using the last 10 historic frames ($N = 10, T = 10$) with 20 DCT coefficients for both the H3.6M and CMU datasets, as well as ($N = 50, T = 10$) with 20 DCT coefficients additionally for H3.6M (here we select only the 20 lowest frequency DCT coefficients). We trained each model for 10 epochs with a batch size of 2048, and a learning rate of 0.00001. The confusion matrices for the H3.6M dataset are shown in figures 3, and 4 respectively. Here, we use the same train set as outlined in section 6.1. However, we report results on subject 11- which for motion prediction was used as the validation set. We did this because the number of instances are much greater than subject 5, and no hyperparameter tuning was necessary. For the CMU dataset we used the same train and test split as for all other experiments.

In both cases, for the H3.6M dataset, the classifier achieves the highest precision score (0.91, 0.95 respectively) for the action *walking* as well as a recall score of 0.83 and 0.81 respectively. Further-

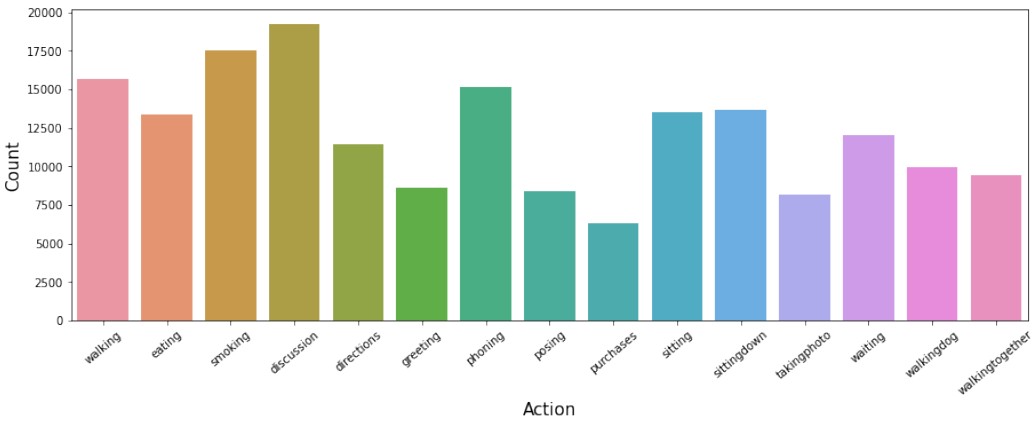

(a) Distribution of short-term training instances for actions in h3.6M.

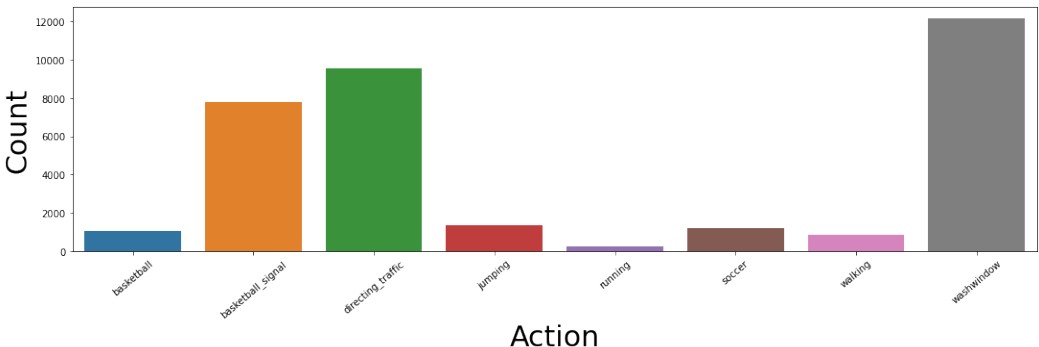

(b) Distribution of training instances for actions in CMU.

Figure 2

more, in both cases *walking together* dominates the false negatives for *walking* (50%, and 44% in each case) as well as the false positives (33% in each case).

The general increase in the distinguishability that can be seen in figure 4 increases the demand to be able to robustly handle distributional shifts as the distribution of values that represent different actions only gets more pronounced as the time scale is increased. This is true with even the näive DCT transformation to capture longer time scales without increasing vector size.

As we can see from the confusion matrix in figure 5, the actions in the CMU dataset are even more easily separable. In particular, our selected ID action in the paper, *Basketball*, can be identified with 100% precision and recall on the test set.

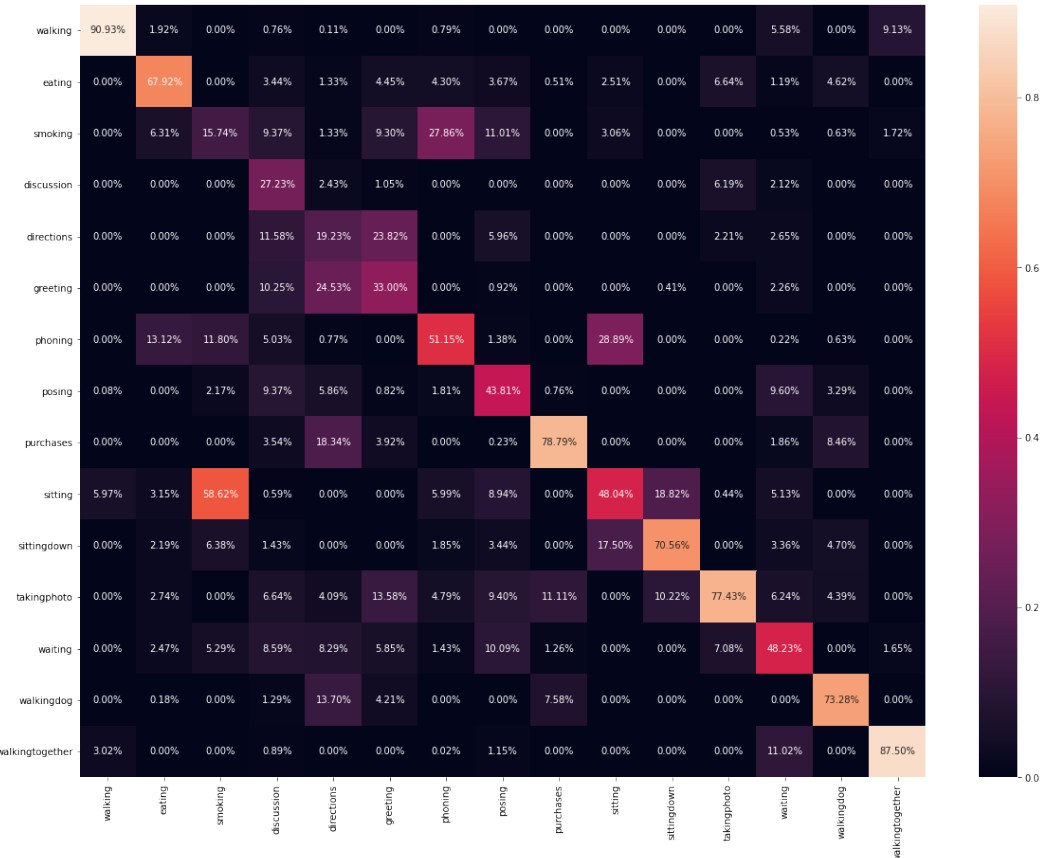

Figure 3: Confusion matrix for a multi-class classifier for action labels. In each case we use the same input convention $\vec{x}_k = [x_{k,1}, \ldots, x_{k,N}, x_{k,N+1}, \ldots, x_{k,N+T}]$ where $x_{k,n} = x_{k,N}$ for $n \geq N$. Such that in each case input to the classifier is $48 \times 20 = 960$. The classifier has 4 fully connected layers. Layer 1: *input dimensions* $\times 1024$, layer 2: $1024 \times 512$, layer 3: $512 \times 128$, layer 4: $128 \times 15$ (or $128 \times 8$ for CMU). Where the final layer uses a softmax to predict the class label. Cross entropy loss is used for training and ReLU activations with a dropout probability of 0.5. We used a batch size of 2048, and a learning rate of $0.00001$.H3.6M dataset. $N = 10$, $T = 10$. Number of DCT coefficients = 20 (lossesless transformation).

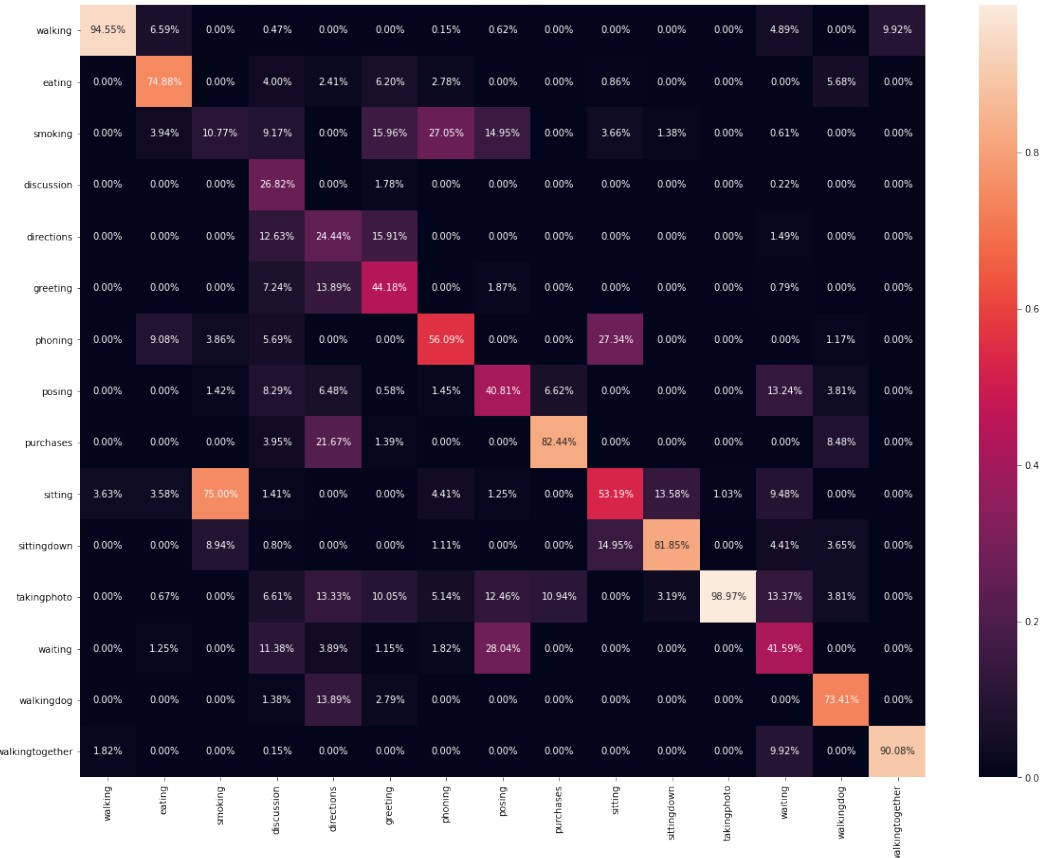

Figure 4: Confusion matrix for a multi-class classifier for action labels. In each case we use the same input convention $\vec{x}_k = [x_{k,1}, \ldots, x_{k,N}, x_{k,N+1}, \ldots, x_{k,N+T}]$ where $x_{k,n} = x_{k,N}$ for $n \geq N$. Such that in each case input to the classifier is $48 \times 20 = 960$. The classifier has 4 fully connected layers. Layer 1: *input dimensions* $\times 1024$, layer 2: $1024 \times 512$, layer 3: $512 \times 128$, layer 4: $128 \times 15$ (or $128 \times 8$ for CMU). Where the final layer uses a softmax to predict the class label. Cross entropy loss is used for training and ReLU activations with a dropout probability of 0.5. We used a batch size of 2048, and a learning rate of $0.00001$. H3.6M dataset. $N = 50$, $T = 10$. Number of DCT coefficients = 20, where the 40 highest frequency DCT coefficients are culled.

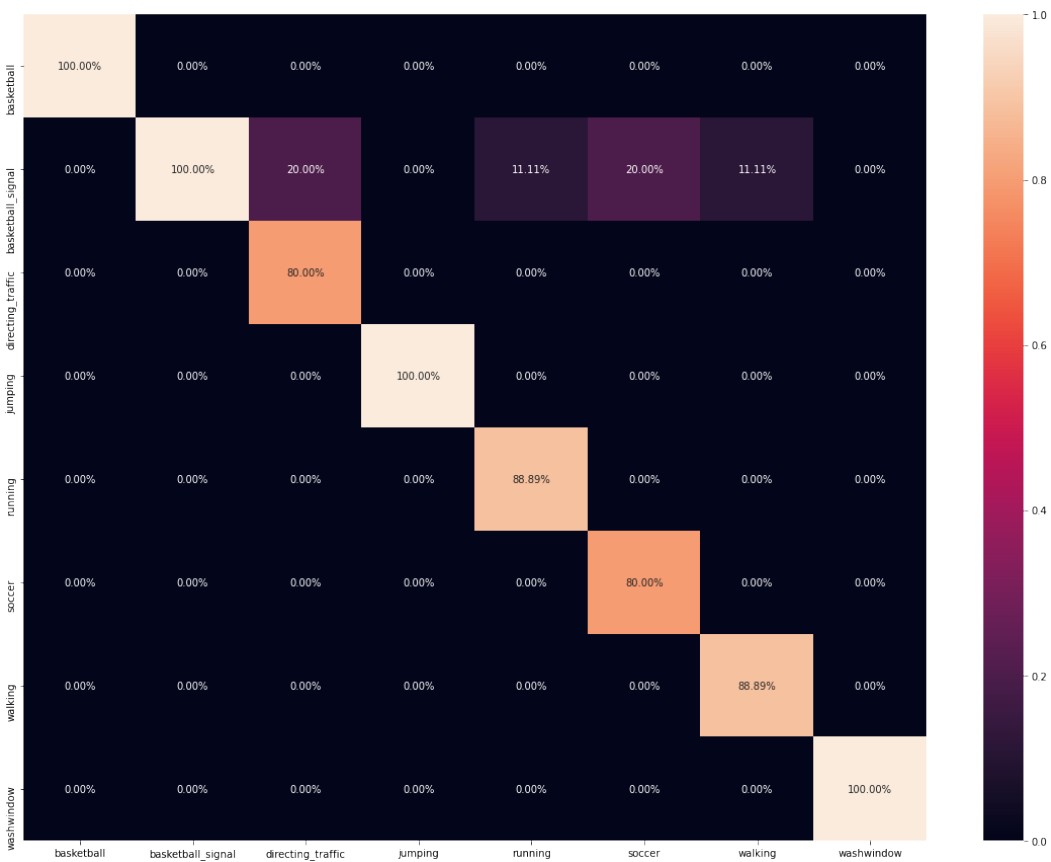

Figure 5: Confusion matrix for a multi-class classifier for action labels. In each case we use the same input convention $\vec{x}_k = [x_{k,1}, \ldots, x_{k,N}, x_{k,N+1}, \ldots, x_{k,N+T}]$ where $x_{k,n} = x_{k,N}$ for $n \geq N$. Such that in each case input to the classifier is $48 \times 20 = 960$. The classifier has 4 fully connected layers. Layer 1: *input dimensions* $\times 1024$, layer 2: $1024 \times 512$, layer 3: $512 \times 128$, layer 4: $128 \times 15$ (or $128 \times 8$ for CMU). Where the final layer uses a softmax to predict the class label. Cross entropy loss is used for training and ReLU activations with a dropout probability of 0.5. We used a batch size of 2048, and a learning rate of 0.00001. CMU dataset. $N = 10$, $T = 25$. Number of DCT coefficients = 35 (lossesless transformation).

# B  FULL RESULTS

| milliseconds | Walking (ID) 80 | 160 | 320 | 400 | Eating (OoD) 80 | 160 | 320 | 400 | Smoking (OoD) 80 | 160 | 320 | 400 | Discussion (OoD) 80 | 160 | 320 | 400 |
|---|---|---|---|---|---|---|---|---|---|---|---|---|---|---|---|---|
| GCN (OoD) | **0.22** | **0.37** | 0.60 | 0.65 | 0.22 | 0.38 | 0.65 | 0.79 | **0.28** | 0.55 | 1.08 | 1.10 | **0.29** | **0.65** | 0.98 | 1.08 |
| Std Dev | 0.001 | 0.008 | . 0.008 | 0.01 | 0.003 | 0.01 | 0.03 | 0.04 | 0.01 | 0.01 | 0.02 | 0.02 | 0.004 | 0.01 | 0.04 | 0.04 |
| ours (OoD) | 0.23 | **0.37** | **0.59** | **0.64** | **0.21** | **0.37** | **0.59** | **0.72** | **0.28** | **0.54** | **1.01** | **0.99** | 0.31 | **0.65** | **0.97** | **1.07** |
| Std Dev | 0.003 | 0.004 | 0.03 | 0.03 | 0.008 | 0.01 | 0.03 | 0.04 | 0.005 | 0.01 | 0.01 | 0.02 | 0.005 | 0.009 | 0.02 | 0.01 |

| milliseconds | Directions (OoD) 80 | 160 | 320 | 400 | Greeting (OoD) 80 | 160 | 320 | 400 | Phoning (OoD) 80 | 160 | 320 | 400 | Posing (OoD) 80 | 160 | 320 | 400 |
|---|---|---|---|---|---|---|---|---|---|---|---|---|---|---|---|---|
| GCN (OoD) | **0.38** | 0.59 | 0.82 | 0.92 | **0.48** | **0.81** | 1.25 | 1.44 | **0.58** | 1.12 | 1.52 | 1.61 | **0.27** | 0.59 | 1.26 | 1.53 |
| Std Dev | 0.01 | 0.03 | 0.05 | 0.06 | 0.006 | 0.01 | 0.02 | 0.02 | 0.006 | 0.01 | 0.01 | 0.01 | 0.01 | 0.05 | 0.1 | 0.1 |
| ours (OoD) | **0.38** | **0.58** | **0.79** | **0.90** | 0.49 | **0.81** | **1.24** | **1.43** | 0.57 | **1.10** | **1.52** | **1.61** | 0.33 | 0.68 | **1.25** | **1.51** |
| Std Dev | 0.007 | 0.02 | 0.0 | 0.05 | 0.006 | 0.005 | 0.02 | 0.02 | 0.004 | 0.003 | 0.01 | 0.01 | 0.02 | 0.05 | 0.03 | 0.03 |

| milliseconds | Purchases (OoD) 80 | 160 | 320 | 400 | Sitting (OoD) 80 | 160 | 320 | 400 | Sitting Down (OoD) 80 | 160 | 320 | 400 | Taking Photo (OoD) 80 | 160 | 320 | 400 |
|---|---|---|---|---|---|---|---|---|---|---|---|---|---|---|---|---|
| GCN (OoD) | **0.62** | 0.90 | 1.34 | 1.42 | 0.40 | 0.66 | 1.15 | 1.33 | 0.46 | 0.94 | 1.52 | 1.69 | **0.26** | 0.53 | 0.82 | **0.93** |
| Std Dev | 0.001 | 0.001 | 0.02 | 0.03 | 0.003 | 0.007 | 0.02 | 0.03 | 0.01 | 0.03 | 0.04 | 0.05 | 0.005 | 0.01 | 0.01 | 0.02 |
| ours (OoD) | **0.62** | **0.89** | **1.23** | **1.31** | **0.39** | **0.63** | **1.05** | **1.20** | **0.40** | **0.79** | **1.19** | **1.33** | **0.26** | **0.52** | **0.81** | 0.95 |
| Std Dev | 0.001 | 0.002 | 0.005 | 0.01 | 0.001 | 0.001 | 0.004 | 0.005 | 0.007 | 0.009 | 0.01 | 0.02 | 0.005 | 0.01 | 0.01 | 0.01 |

| milliseconds | Waiting (OoD) 80 | 160 | 320 | 400 | Walking Dog (OoD) 80 | 160 | 320 | 400 | Walking Together (OoD) 80 | 160 | 320 | 400 | Average (of 14 for OoD) 80 | 160 | 320 | 400 |
|---|---|---|---|---|---|---|---|---|---|---|---|---|---|---|---|---|
| GCN (OoD) | **0.29** | 0.59 | **1.06** | 1.30 | **0.52** | **0.86** | 1.18 | **1.33** | 0.21 | 0.44 | 0.67 | 0.72 | **0.38** | 0.69 | 1.09 | 1.27 |
| Std Dev | 0.01 | 0.03 | 0.05 | 0.05 | 0.01 | 0.02 | 0.02 | 0.03 | 0.005 | 0.02 | 0.03 | 0.03 | 0.007 | 0.02 | 0.04 | 0.04 |
| ours (OoD) | **0.29** | **0.58** | **1.06** | **1.29** | **0.52** | 0.88 | **1.17** | 1.34 | 0.21 | 0.44 | 0.66 | 0.74 | **0.38** | 0.68 | 1.07 | 1.21 |
| Std Dev | 0.0007 | 0.003 | 0.001 | 0.006 | 0.006 | 0.01 | 0.008 | 0.01 | 0.01 | 0.01 | 0.01 | 0.01 | **0.006** | **0.01** | **0.01** | **0.02** |

Table 6: Short-term prediction of Eucildean distance between predicted and ground truth joint angles on H3.6M. Each experiment conducted 3 times. We report the mean and standard deviation. Note that we have lower variance in results for ours.

| milliseconds | Basketball (ID) 80 | 160 | 320 | 400 | 1000 | Basketball Signal (OoD) 80 | 160 | 320 | 400 | 1000 | Directing Traffic (OoD) 80 | 160 | 320 | 400 | 1000 |
|---|---|---|---|---|---|---|---|---|---|---|---|---|---|---|---|
| GCN | **0.40** | 0.67 | **1.11** | **1.25** | **1.63** | **0.27** | **0.55** | **1.14** | **1.42** | 2.18 | 0.31 | 0.62 | 1.05 | 1.24 | 2.49 |
| ours | **0.40** | **0.66** | 1.12 | 1.29 | 1.76 | 0.28 | 0.57 | 1.15 | 1.43 | **2.07** | **0.28** | **0.56** | **0.96** | **1.10** | **2.33** |

| milliseconds | Jumping (OoD) 80 | 160 | 320 | 400 | 1000 | Running (OoD) 80 | 160 | 320 | 400 | 1000 | Soccer (OoD) 80 | 160 | 320 | 400 | 1000 |
|---|---|---|---|---|---|---|---|---|---|---|---|---|---|---|---|
| GCN | 0.42 | 0.73 | **1.72** | **1.98** | **2.66** | **0.46** | 0.84 | 1.50 | 1.72 | **1.57** | 0.29 | 0.54 | 1.15 | 1.41 | 2.14 |
| ours | **0.38** | **0.72** | 1.74 | 2.03 | 2.70 | **0.46** | **0.81** | **1.36** | **1.53** | 2.09 | **0.28** | **0.53** | **1.07** | **1.27** | **1.99** |

| milliseconds | Walking (OoD) 80 | 160 | 320 | 400 | 1000 | Washing window (OoD) 80 | 160 | 320 | 400 | 1000 | Average (of 7 for OoD) 80 | 160 | 320 | 400 | 1000 |
|---|---|---|---|---|---|---|---|---|---|---|---|---|---|---|---|
| GCN | 0.40 | 0.61 | 0.97 | 1.18 | 1.85 | 0.36 | 0.65 | 1.23 | **1.51** | 2.31 | 0.36 | 0.65 | 1.41 | 1.49 | 2.17 |
| ours | **0.38** | **0.54** | **0.82** | **0.99** | **1.27** | **0.35** | **0.63** | **1.20** | **1.51** | 2.26 | **0.34** | **0.62** | **1.35** | **1.41** | **2.10** |

Table 7: Eucildean distance between predicted and ground truth joint angles on CMU. Full table.

| milliseconds | Basketball (ID) 80 | 160 | 320 | 400 | 1000 | Basketball Signal (OoD) 80 | 160 | 320 | 400 | 1000 | Directing Traffic (OoD) 80 | 160 | 320 | 400 | 1000 |
|---|---|---|---|---|---|---|---|---|---|---|---|---|---|---|---|
| GCN | **15.7** | **28.9** | **54.1** | **65.4** | 108.4 | 14.4 | 30.4 | 63.5 | 78.7 | 114.8 | 18.5 | 37.4 | **75.6** | **93.6** | 210.7 |
| ours | 16.0 | 30.0 | 54.5 | 65.5 | **98.1** | **12.8** | **26.0** | **53.7** | **67.6** | **103.2** | **18.3** | **37.2** | 75.7 | 93.8 | **199.6** |

| milliseconds | Jumping (OoD) 80 | 160 | 320 | 400 | 1000 | Running (OoD) 80 | 160 | 320 | 400 | 1000 | Soccer (OoD) 80 | 160 | 320 | 400 | 1000 |
|---|---|---|---|---|---|---|---|---|---|---|---|---|---|---|---|
| GCN | **24.6** | **51.2** | 111.4 | 139.6 | 219.7 | 32.3 | 54.8 | 85.9 | 99.3 | **99.9** | 22.6 | 46.6 | 92.8 | 114.3 | **192.5** |
| ours | 25.0 | 52.0 | **110.3** | **136.8** | **200.2** | **29.8** | **50.2** | **83.5** | **98.7** | 107.3 | **21.1** | **44.2** | **90.4** | **112.1** | 202.0 |

| milliseconds | Walking (OoD) 80 | 160 | 320 | 400 | 1000 | Washing window (OoD) 80 | 160 | 320 | 400 | 1000 | Average of 7 for (OoD) 80 | 160 | 320 | 400 | 1000 |
|---|---|---|---|---|---|---|---|---|---|---|---|---|---|---|---|
| GCN | 10.8 | 20.7 | 42.9 | 53.4 | 86.5 | **17.1** | **36.4** | **77.6** | **96.0** | **151.6** | **20.0** | 43.8 | 86.3 | 105.8 | 169.2 |
| ours | **10.5** | **18.9** | **39.2** | **48.6** | **72.2** | 17.6 | 37.3 | 82.0 | 103.4 | 167.5 | 21.6 | **42.3** | **84.2** | **103.8** | **164.3** |

Table 8: Mean Joint Per Position Error (MPJPE) between predicted and ground truth 3D Cartesian coordinates of joints on CMU. Full table.

| | Walking (ID) | | | | Eating (OoD) | | | | Smoking (OoD) | | | | Discussion (OoD) | | | |
|---|---|---|---|---|---|---|---|---|---|---|---|---|---|---|---|---|
| milliseconds | 560 | 720 | 880 | 1000 | 560 | 720 | 880 | 1000 | 560 | 720 | 880 | 1000 | 560 | 720 | 880 | 1000 |
| attention-GCN (OoD) | **55.4** | **60.5** | **65.2** | **68.7** | 87.6 | 103.6 | 113.2 | 120.3 | 81.7 | 93.7 | 102.9 | 108.7 | **114.6** | 130.0 | **133.5** | **136.3** |
| ours (OoD) | 58.7 | 60.6 | 65.5 | 69.1 | **81.7** | **94.4** | **102.7** | **109.3** | **80.6** | **89.9** | **99.2** | **104.1** | 115.4 | **129.0** | 134.5 | 139.4 |
| | Directions (OoD) | | | | Greeting (OoD) | | | | Phoning (OoD) | | | | Posing (OoD) | | | |
| milliseconds | 560 | 720 | 880 | 1000 | 560 | 720 | 880 | 1000 | 560 | 720 | 880 | 1000 | 560 | 720 | 880 | 1000 |
| attention-GCN (OoD) | **107.0** | 123.6 | 132.7 | 138.4 | **127.4** | 142.0 | 153.4 | 158.6 | 98.7 | 117.3 | 129.9 | 138.4 | **151.0** | **176.0** | **189.4** | **199.6** |
| ours (OoD) | 107.1 | **120.6** | **129.2** | **136.6** | 128.0 | **140.3** | **150.8** | **155.7** | **95.8** | **111.0** | **122.7** | **131.4** | 158.7 | 181.3 | 194.4 | 203.4 |
| | Purchases (OoD) | | | | Sitting (OoD) | | | | Sitting Down (OoD) | | | | Taking Photo (OoD) | | | |
| milliseconds | 560 | 720 | 880 | 1000 | 560 | 720 | 880 | 1000 | 560 | 720 | 880 | 1000 | 560 | 720 | 880 | 1000 |
| attention-GCN (OoD) | **126.6** | 144.0 | **154.3** | **162.1** | **118.3** | 141.1 | 154.6 | 164.0 | **136.8** | 162.3 | 177.7 | 189.9 | **113.7** | 137.2 | 149.7 | 159.9 |
| ours (OoD) | 128.0 | **143.2** | 154.7 | 164.3 | 118.4 | **137.7** | **149.7** | **157.5** | **136.8** | 157.6 | 170.8 | 180.4 | 116.3 | **134.5** | **145.6** | **155.4** |
| | Waiting (OoD) | | | | Walking Dog (OoD) | | | | Walking Together (OoD) | | | | Average (of 14 for OoD) | | | |
| milliseconds | 560 | 720 | 880 | 1000 | 560 | 720 | 880 | 1000 | 560 | 720 | 880 | 1000 | 560 | 720 | 880 | 1000 |
| attention-GCN (OoD) | **109.9** | 125.1 | 135.3 | 141.2 | **131.3** | **146.9** | **161.1** | **171.4** | **64.5** | **71.1** | **76.8** | **80.8** | **112.1** | 129.6 | 140.3 | 147.8 |
| ours (OoD) | 110.4 | **124.5** | **133.9** | **140.3** | 138.3 | 151.2 | 165.0 | 175.5 | 67.7 | 71.9 | 77.1 | **80.8** | 113.1 | **127.7** | **137.9** | **145.3** |

Table 9: Long-term prediction of 3D joint positions on H3.6M. Here, ours is also trained with the attention-GCN model.

## C    LATENT SPACE OF THE VAE

One of the advantages of having a generative model involved is that we have a latent variable which represents a distribution over deterministic encodings of the data. We considered the question of whether or not the VAE was learning anything interpretable with its latent variable as was the case in Kipf & Welling (2016).

The purpose of this investigation was two-fold. First to determine if the generative model was learning a comprehensive internal state, or just a non-linear average state as is common to see in the training of VAE like architectures. The result of this should suggest a key direction of future work. Second, an interpretable latent space may be of paramount usefulness for future applications of human motion prediction. Namely, if dimensionality reduction of the latent space to an inspectable number of dimensions yields actions, or behaviour that are close together if kinematically or teleolgically similar, as in Bourached & Nachev (2019), then human experts may find unbounded potential application for a interpretation that is both quantifiable and qualitatively comparable to all other classes within their domain of interest. For example, a medical doctor may consider a patient to have unusual symptoms for condition, say, A. It may be useful to know that the patient's deviation from a classical case of A, is in the direction of condition, say, B.

We trained the augmented GCN model discussed in the main text with all actions, for both datasets. We use Uniform Manifold Approximation and Projection (UMAP) (McInnes et al., 2018) to project the latent space of the trained GCN models onto 2 dimensions for all samples in the dataset for each dataset independently. From figure 6 we can see that for both models the 2D project relatively closely resembles a spherical gaussian. Further, we can see from figure 6b that the action *walking* does not occupy a discernible domain of the latent space. This result is further verified by using the same classifier as used in appendix A, which achieved no better than chance when using the latent variables as input rather than the raw data input.

This result implies that the benefit observed in the main text is by using the generative model is significant even if the generative model has poor performance itself. In this case we can be sure that the reconstructions are at least not good enough to distinguish between actions. It is hence natural for future work to investigate if the improvement on OoD performance is greater if trained in such a way as to ensure that the generative model performs well. There are multiple avenues through which such an objective might be achieve. Pre-training the generative model being one of the salient candidates.

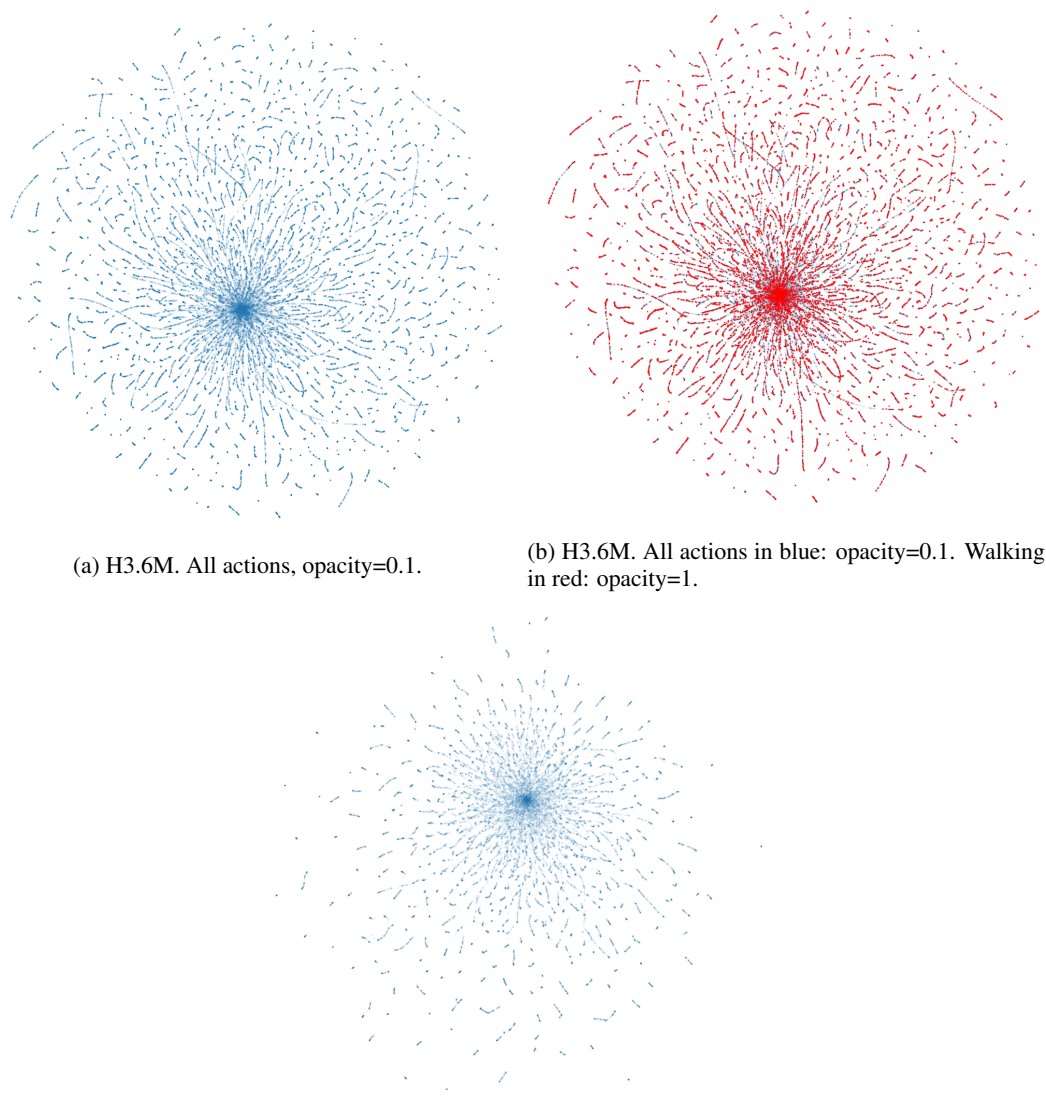

(a) H3.6M. All actions, opacity=0.1.

(b) H3.6M. All actions in blue: opacity=0.1. Walking in red: opacity=1.

(c) CMU. All actions in blue: opacity=0.1.

Figure 6: Latent embedding of the trained model on both the H3.6m and the CMU datasets independently projected in 2D using UMAP from 384 dimensions for H3.6M, and 512 dimensions for CMU using default hyperparameters for UMAP.

# D  ARCHITECTURE DIAGRAMS

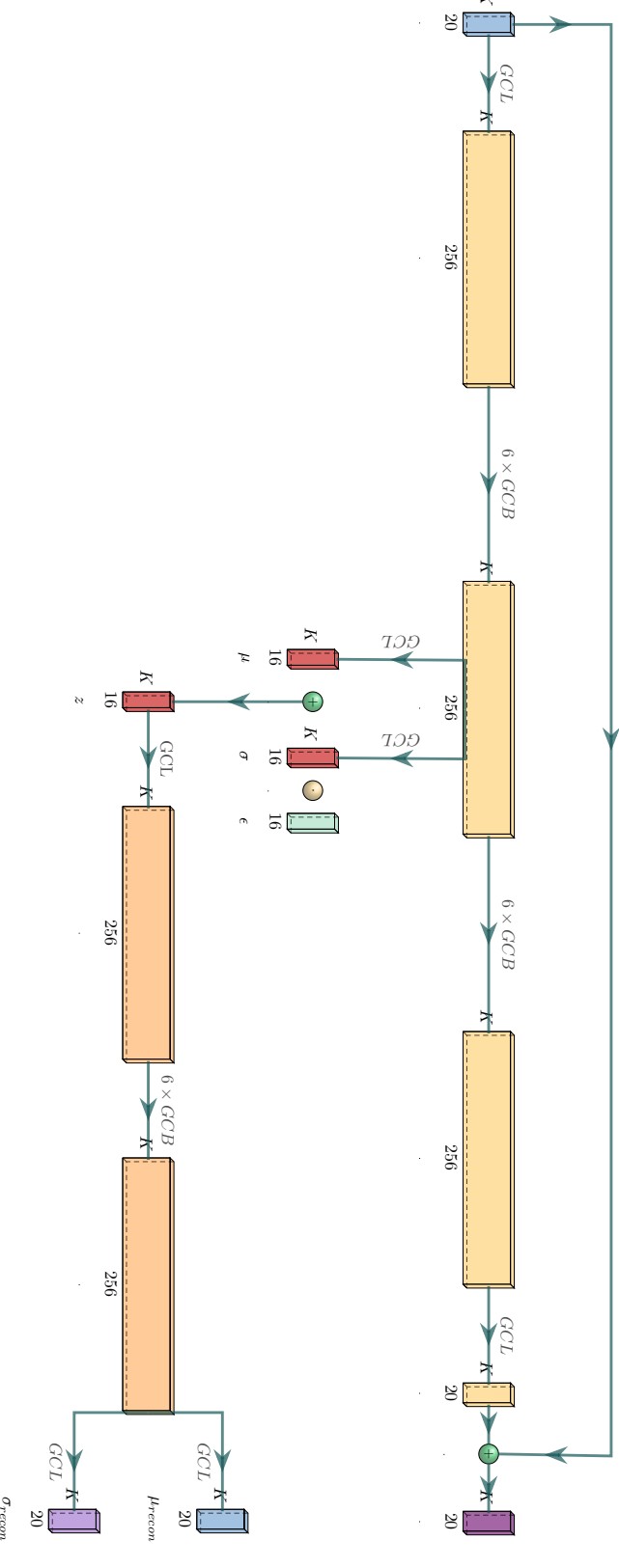

Figure 7: Network architecture with discriminative and VAE branch.

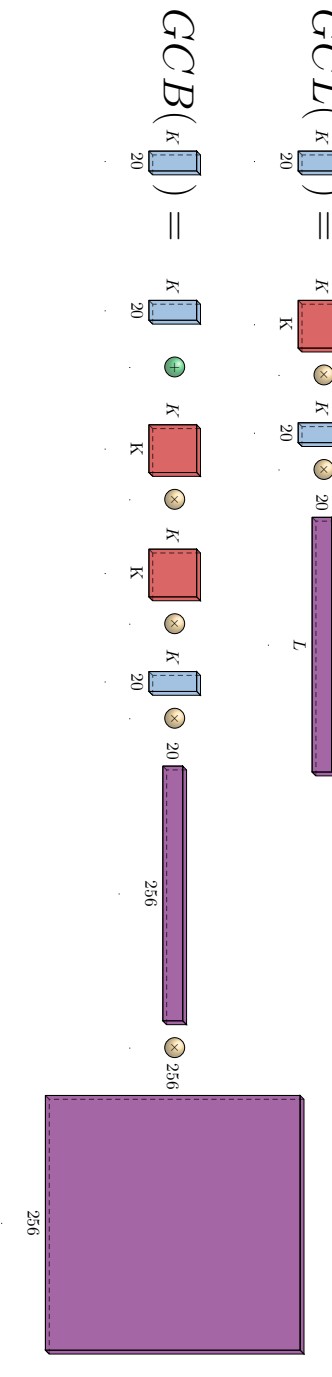

Figure 8: Graph Convolutional layer (GCL) and a residual graph convolutional block (GCB).

