# OpenReview forum: "GENERATIVE MODEL-ENHANCED HUMAN MOTION PREDICTION"
_ICLR.cc/2021/Conference — Reject_

### Official Review · AnonReviewer3 · 2020-10-28
**Review - Generative Model-Enhanced Human Motion Predicition**

**Rating:** 3
**Confidence:** 5

**Review:**

## Summary:
---
This paper raises and studies concerns about the generalization of 3D human motion prediction approaches across unseen motion categories. The authors address this problem by augmenting existing architectures with a VAE framework. More precisely, an encoder network that is responsible for summarizing the seed sequence is shared by two decoders for the reconstruction of the seed motion and prediction of the future motion. Hence, the encoder is trained by using both the ELBO of a VAE and the objective of the original motion prediction task.

## Pros:
---
The paper has a novel and interesting direction as robustness to distribution shifts has not been studied before in 3D human motion modeling. It is implemented around one of the SoTA models based on Graph Convolutional Networks (GCN) using discrete cosine transformation (DCT) features extracted from the motion sequence. To simulate the out-of-distribution scenario, the baselines and the proposed extension are trained on a single action category such as walking and tested on the remaining actions such as eating and sitting. Experiment results on the H3.6M and CMU datasets show that the proposed approach is useful on out-of-distribution (OoD) test cases.

## Cons:
---
I have two main concerns on the proposed benchmark and the models.

-- OoD Benchmark--
- It looks like there is a significant underfitting problem. The performance of GCN on the walking category is 0.56 at 400 ms while the in-distribution (ID) performance with OoD training is 0.66 (Table 1). The training split for the OoD setup proposed by the authors is possibly too small. I also can not grasp the motivation for selecting a training set “as small in quantity, and narrow in domain as possible” (Section 3). While there is not enough or barely enough training samples, the comparisons might be misleading. We do not know how the proposed extension behaves on the standard task. The authors should compare their models on the main task as well.

- Motion samples from different categories (i.e., walking, eating, etc.) can still be useful for the models in learning the 3D human motion prior. In fact, it has been shown by Martinez et al. (2017) [4] that training motion models with _all_ available actions improves the performance significantly compared to a single-action models as done in this paper. While the proposed approach outperforms the baselines in average performance, it is not always or substantially better on the fine-grained actions.

- It is a tedious setup, but a leave-one-action-out strategy can be more reliable. In my opinion a better option would be training on one dataset and testing on another one. This would allow for an evaluation of the existing (and even pre-trained) models directly where the proposed extension would remain as the only factor for evaluation. In the context of H3.6M and CMU datasets, this might not be straightforward due to different skeletal configurations. Yet there exists a much larger benchmark for 3D motion prediction: AMASS [1]. This would be a suitable candidate for this task as it is a collection of several diverse mocap datasets with different motion categories. It would be very easy to train on a subset of datasets and test on the remaining ones as all the datasets follow a unified skeletal configuration. Note that this is only a suggestion to improve the current work and I am not asking for running experiments on AMASS for the rebuttal as it would drastically change the submission.

-- Quantifying the OoD--
- The existing architectures are augmented with a VAE latent space and a decoder, which is not technically novel. However, regularization of the representation space and reconstruction of the inputs as auxiliary tasks seem like helpful to the motion prediction task and a good contribution under the “limited” training/evaluation protocol. At the same, time the proposed evaluation protocol is not orthogonal to the task but instead it just follows the existing task protocol. In other words, it is not an independent metric/method/framework for assessing the existing models’ OoD performance.

I am asking the following questions as the proposed approach is presented as a “framework”:

- The authors hypothesize that motion prediction in generative modeling frameworks can alleviate the OoD problems. I find it too broad as generative modelling can be applied in various frameworks. Although it is conceptually very different, [1] uses an auto-regressive model, which is a generative model by design, and trains the model by predicting both the seed (i.e., loosely reconstruction) and the future frames similar to the proposed approach. Can we say that they also deal with the OoD problems implicitly? How do the authors position their “framework” compared to this line of work?

- The authors only focus on the Seq2seq-based methods for motion prediction and choose a baseline with an implicit temporal model (i.e., using DCT to encode the motion sequences). Hence, the proposed approach seems to be limited to this GCN-based architecture. How can the sequential deterministic models [1, 3, 4] be addressed?

-- Additional comments --
- Figures should be improved. Especially the text is hardly readable.

- I find it very hard to follow Section 3. It was clear only after I read the section A in the appendix. It would be clearer if some of the findings are discussed in Section 3 already.

- The losses in the tables are too high compared to the actual task. I am not sure if there is a qualitative difference between the models as the authors did not present any qualitative results.

- Missing related work on 3D motion modelling. I list a s,all collection of SoA representatives below:

[1] Aksan, Emre, Manuel Kaufmann, and Otmar Hilliges. "Structured prediction helps 3d human motion modelling." Proceedings of the IEEE International Conference on Computer Vision. 2019.
[2] Gui, L. Y., Wang, Y. X., Liang, X., & Moura, J. M. (2018). Adversarial geometry-aware human motion prediction. In Proceedings of the European Conference on Computer Vision (ECCV) (pp. 786-803).
[3] Pavllo, D., Grangier, D., & Auli, M. (2018). Quaternet: A quaternion-based recurrent model for human motion. arXiv preprint arXiv:1805.06485.
[4] Martinez, J., Black, M. J., & Romero, J. (2017). On human motion prediction using recurrent neural networks. In Proceedings of the IEEE Conference on Computer Vision and Pattern Recognition (pp. 2891-2900).

---

> ### Author Response · Authors · 2020-11-23
> **We are grateful for the reviewer’s comments, which we address in detail below.**
>
> -- OoD Benchmark--
> We disagree. Any good benchmark should seek to replicate the circumstances and demands of the real-world
> task while being maximally sensitive to the performance differences between competing models. The critical point
> we make at the outset of our paper is that the first property is very difficult to satisfy owing to the extreme
> heterogeneity and complex compositionality of human actions, and interacts with the more important second
> property. Any modest selection of classes is likely to contain a blend of degrees of similarity, so that the extent to
> which any one class is out-of-sample with respect to the set will vary a great deal. When a set of models are
> evaluated within a leave-one-class-out framework, performance will then be brittle, dependent on the accidental
> homology between the test class and one or more of the training classes. Crucially, performance in such
> circumstances will be less sensitive to the generalising power of the test model than when evaluated within the
> converse framework—testing on all after training on one—for generalisation will be easier in being informed by a
> greater diversity of training data. The latter framework helpfully accentuates the difference between the training
> and testing data, and broadens the range of contrasts being evaluated. Equally, given that an action of a given
> distinctive morphology might be relatively rare, data efficiency is an important concern here, and ought to be
> stress-tested by varying the quantity of training data as well as its composition.
>
> We agree that the AMASS dataset would be a good candidate for further benchmarks for OoD motion, but a
> comparison across different sets of instances of supposedly the same action is not the critical contrast, as we
> have already argued. What we need is a comparison across explicitly labelled actions, for the fundamentals of
> real-world action render all performance subject to unquantifiable and likely substantial distributional shifts. This
> is what our benchmark sets out to achieve.
>
> -- Quantifying the OoD--
> The protocol is not the same: we modify it so as to quantify the difference between in- and out-of-distribution
> performance for two models that differ only in one of them being enhanced with a generative model. The same
> approach can be taken for other models in other contexts.
>
> On the reviewers questions:
> - On comparison to Askan et al.
> The cardinal characteristics of human action we draw attention to in the introduction suggest deep generative
> architectures are likely to provide the best means of modelling it. But the focus of this paper is the felicity of the
> simpler, often computationally more economical, approach of enhancing a conventional discriminative model with
> generative machinery. We call it a framework because it is transferable across an array of discriminative
> architectures, at least when implemented with neural networks of sufficient flexibility. That others may achieve
> hardening to OoD problems via kindred mechanisms—in Aksan et al.’s work the reviewer cites via explicit
> probabilistic modelling of joint dependence—seems to us to reinforce the general point we are making here, and
> to justify the introduction of the OoD benchmark I think we are all agreed is overdue.
>
> - On how can sequential deterministic models be addressed:
> While our focus is on the current SOTA model, which happens to be a GCN model, the approach is potentially
> applicable to any discriminative model flexibly implemented as a neural network. Weight sharing with an auxiliary,
> subservient generative model will always be possible, and would encourage the discriminative model's sequence
> to contain a richer description of the input. Additionally, an auxiliary generative model that generates a sequence
> in tandem with, but separate from, the discriminative model could share its uncertainty measures and alternative
> futures at each point in the generation of the sequence. Whereas in our case we encourage the data flowing
> through a discriminative model to double as the representation in a latent variable model, in a sequential
> deterministic model, a fully or partially generated sequence could similarly serve a dual role as the seed for a
> generative sequential model’s sequence.
>
> How these tricks are used very much depends on the experimental setup, and while it is natural to consider the
> analogous approaches that may be applicable to sequential models, the considerations and metrics relevant to
> the juxtaposition of feedforward and sequential models are not. We believe that to include this investigation would
> increase the complexity of this paper to the point of obfuscating its main contribution.
>
> -- Additional comments --
> Amended as requested and may now be viewed in the pdf. Note we cite Martinez et al., already, but have added others: many thanks.

---

### Official Review · AnonReviewer4 · 2020-10-28
**Review of "GENERATIVE MODEL-ENHANCED HUMAN MOTION PREDICTION"**

**Rating:** 4
**Confidence:** 3

**Review:**

The paper presents, firstly, a new benchmark (based on Human3.6M and CMU datasets) for human activity and motion with a high degree of out-of-distribution examples, and secondly a hybrid framework for human motion prediction which is more robust to out-of-distribution samples.

On a positive note, the presented view of human activity as highly compositional and without a clear ontology of actions and sub-actions is highly relevant, and the observed issues with the state of the art methods are completely correctly characterized. The proposed behchmark is also highly valuable to the community.

However, the paper suffer from two flaws that renders it unfit for publication in ICLR in its current form.
* Firstly, the contribution - to combine GCN with the approach of (Myronenko 2018) to regularize the training with a generative model that takes unlabeled samples into accound, and to replace their VAE framework with a corresponding VGAE one -  is not on its own significant enough to serve as a basis for an ICLR paper.
* Secondly, the experiments are not adequate in that the method is only compared to a GCN without the generative model - and not with any of the other state-of-the-art in motion prediction. Moreover, results are presented without standard deviations which makes it hard to determine if the improvements are significant.

These two flaws leads to a Reject recommendation, but the authors are highly encouraged to expand the experiments to empirically verify that the proposed contribution is significant enough to warrant publication, and resubmit to a later conference. While addressing the second flaw, it would also help convince the reader of the significance of the method contribution.

---

> ### Author Response · Authors · 2020-11-23
> **We are grateful for the reviewer’s comments, which we address in detail below.**
>
> On the contribution: The judgment of publication significance is a subjective, editorial matter. But the reviewer states our contribution only partially. The value of our work is less in the specific implementation of a generatively-enhanced model of
> motion—though the implementation is novel—than in the demonstration of the general value of the approach in
> the domain of action modelling. We do this for reasons the reviewer agrees with—the complex compositionality
> and constitutional indeterminacy of human motion—which have not been adequately discussed in the literature,
> and have a bearing on all models of human action. The lack of sufficient awareness of the problem is reflected in
> the absence of an established OoD benchmark, which we provide here: another aspect of our contribution the
> reviewer describes as highly valuable.
>
> On the experiments: Our objective is to demonstrate the value of enhancing a discriminative model with a generative one: the correct comparison is therefore with the same model, but without the generative machinery. We have chosen the state-
> of-the-art model at the time of submission: space precludes evaluation of a range of models, but demonstrating
> an effect on less successful models would naturally be weaker, for they leave more room for improvement. In
> reporting model performance we follow established practice in the literature, but have now conducted additional experiments to provide confidence intervals for the h3.6m dataset results. Which may now be viewed in the pdf.

---

### Official Review · AnonReviewer2 · 2020-10-29
**The proposed approach is reasonalbe for dealing with Out-of-Distribution (OoD) problem of human motion prediction. But the experimental results are unconvinced.**

**Rating:** 5
**Confidence:** 4

**Review:**

This paper presents a generative model to solve the OoD problem in human motion prediction. It extends the GCN and attention-GCN works with VGAE for predicting human motions that are  different from ones used in training. Experiments are performed on H36M and CMU benchmarks for illustrating the efficacy of the proposed approach.

Pros:
1. The paper is good in writting and easy to follow the idea
2. The perspective of using generative model to deal with OoD problem in human motion is novel.

Cons.
1. My major concern is the effectiveness of the proposed approach. From the results shown in Table 3 to Table 5, we could find that the proposed approach fail to solve the OoD problem for some actions when comparing with the baseline attention-GCN. For example, in Table 5, the proposed generative model achieve poor performance than attention-GCN, such as Discussion, Posing, Purchases, Walking Dog and Walking Together (5 out of 14 acitions). These experiments could not provide convinced results to depict the efficacy of the proposed approach. I think the authors should provide more explanations on this which should make this paper stronger.

---

> ### Author Response · Authors · 2020-11-23
> **We are grateful for the reviewer’s comments, which we address in detail below.**
>
> The results show that our approach matches or exceeds the current state-of-the-art in individual tasks, and
> exceeds the state-of-the-art overall. A choice between the two architectures of similar size and training
> characteristics would naturally find in favour of ours, for that is what the evaluation data compel. But our aim here
> is less to build a state-of-the-art model of motion prediction than to show that the use of a relatively simple
> generative model to enhance a discriminative model of a radically different architecture improves OoD
> performance even when the discriminative architecture is already heavily optimised. It is the felicity of that simple
> general architectural move that we wish to highlight here, for it has implications for the design of other models in
> the field, indeed any model deployed on the same task. We have amended the text to make this clear.
>
> We have also conducted additional experiments to provide confidence intervals for the h3.6m dataset results. Which may now be viewed in the pdf.

---

### Official Review · AnonReviewer1 · 2020-10-29

**Rating:** 5
**Confidence:** 4

**Review:**

Summary:
This paper proposes a method and benchmark for out-of-distribution modeling and evaluation of human motion. They evaluate against state-of-the-art human motion methods, and show favorable performance against them.


Pros
+ Generative model formulation for human motion prediction
+ Benchmark for testing out of distribution performance in Human 3.6M and CMU-Mocap
+ Proposed generative model outperforms baselines


Comments / Suggestions:
- Interpretability claim:
The authors talk about facilitating interpretability in the abstract, however, I fail to find any clear experiments suggesting this. For example, I cannot find analysis of the different dimensions in the learned latent space or anything of that nature. I see section B in the supplementary material discusses interpretability, but I fail to find any clear cut results about this. Can the authors clarify how this claim is reflected in the paper?

- Evaluations:
The evaluations provided in this paper are based on euclidean distance measured with respect to the ground truth. While this metric is reasonable, it may also not be enough to evaluate a generative model of motion (e.g., there are multiple plausible futures given a single past). Given that there are clearly defined actions in the used datasets, I would suggest using a metric that measures the generated sequences as a whole. For example, one can train a motion recognition network which given a motion tells us what type of motion we are observing. The authors could train this type of network and test it on their generated motion to see if the predicted / generated motion is recognized as the right category. Another similar evaluation would be FID, where the authors can see if the predicted / generated motion distribution in feature space is close to the ground truth distribution.

- Differences with Kipf & Welling, 2016:
The authors mention that they adopt VGAE from Kipf and Welling, but I fail to find where the authors mention what are the specific differences of their method in comparison to Kipf & Welling, 2016. Can the authors clarify this or point out where the specific differences are mentioned?


Conclusion:
The proposed benchmark is interesting and useful for out-of-distribution evaluations, however, some evaluations may be missing to make this more comprehensive. The differences between the method used by the authors and the related work need to be clarified. I am willing to change my score if the authors successfully address the issues mentioned above.

###########################
  Post Rebuttal Comments
###########################

After reading the rebuttal, I am keeping my original score. For my first concern, they authors mention space as being a limitation for not providing analysis on the "surveyable" latent space, but as far as I know, additional experiments addressing my concern could have been added to the supplementary material. For my second concern, the authors talk about excelling in synthetic fidelity, however, there are fidelity measures for generative models that were not used in this submission. MSE is not a fidelity metric. I suggest that the authors address the concerns raised by the reviewers in future submissions, and it's highly likely that the work will be more solid.

---

> ### Author Response · Authors · 2020-11-23
> **We are grateful for the reviewer’s comments, which we address in detail below.**
>
> Interpretability: Our point is that the introduction of a succinct, surveyable latent space can facilitate the identification of
> characteristic patterns of motion variability by rendering them intuitively apprehensible. This is a theoretical claim
> implied by the fundamentals of the approach, and here we merely draw attention to an additional benefit our
> approach could potentially bring. Limited space precludes detailed empirical exploration of its value. We have
> amended the text to clarify this point.
>
> Evaluations: Our objective is to demonstrate that a state-of-the-art discriminative predictive model of motion can be hardened
> to OOD challenge by the addition of a relatively simple generative network. The primary measure of performance
> must therefore be the fidelity of motion prediction—not action synthesis or action recognition—for that is our task.
> A generative model that excels in synthetic fidelity and disentanglement of the constituent actions is theoretically
> likely to perform well, but since the device is here deployed subserviently to a discriminative model, the
> appropriate metric is that of the target model it serves to enhance. Indeed, Myronenko’s work suggests a
> relatively crude generative model of limited synthetic power can nonetheless promote a basic discriminative
> architecture to state-of-the-art. What seems to us striking here is that the addition to discriminative architectures
> of fairly simple generative machinery can be remarkably useful, achieving state-of-the-art performance without
> the architectural complexities and training demands an accomplished generative model involves. We have
> amended the text to clarify this point.
>
> Differences with Kipf & Welling, 2016: Our generative model is a variational autoencoder (Kingma & Welling) with graph convolutional layers in place of dense layers, except for immediately around the (dense) layers that produce a sample from q(z|x). Kipf & Welling's application is a link prediction task in citation networks and thus it is desired to model only connectivity in the latent space. Here we model connectivity, position, and temporal frequency. We have amended the text to clarify this.
>
> Thank you: we hope our replies are satisfactory.

---

### Decision · Program_Chairs · 2021-01-07
**Final Decision**

**Decision:**

Reject

**Comment:**

This paper proposes a method for out-of-distribution modeling and evaluation in  the human motion prediction task. Paper was reviewed by four expert reviewers who identified the following pros and cons.

> Pros:
- New benchmark for testing out of distribution performance [R1]
- Compelling performance with respect to the baselines [R1,R4]
- Paper is well written and easy to  follow  [R2]
- Generative model in the context  of  out-of-distribution modeling of human motion is novel [R1,R2,R4]

> Cons:
- Lack of support for interpretability claim  [R1]
- Validity and usefulness of the metric [R1]
- Lack of "effectiveness" of the proposed approach [R2,R4]
- Technical contributions are not significant [R3,R4]
- Experimental validation lacks comparisons to other state-of-the-art in motion prediction  methods [R3]
- Lack of evaluation on additional datasets and for the main task [R4]

Authors tried to address the comments in the rebuttal, but  largely unconvincingly to the  reviewers.  On balance, reviewers felt that negatives outweighed the positives and unanimously suggest rejection. AC concurs and sees no reason to overturn this consensus.